# Pumpkin, Cauliflower and Broccoli as New Carriers of Thiamine Compounds for Food Fortification

**DOI:** 10.3390/foods10030578

**Published:** 2021-03-10

**Authors:** Krystyna Szymandera-Buszka, Justyna Piechocka, Agata Zaremba, Monika Przeor, Anna Jędrusek-Golińska

**Affiliations:** Department of Gastronomy Science and Functional Foods, Faculty of Food Science and Nutrition, Poznań University of Life Sciences, Wojska Polskiego 31, 61-624 Poznan, Poland; justyna.piechocka@up.poznan.pl (J.P.); agata.zaremba@up.poznan.pl (A.Z.); monika.przeor@up.poznan.pl (M.P.); annajedrusek-golinska@up.poznan.pl (A.J.-G.)

**Keywords:** thiamine, thiamine carriers, pumpkin, vegetable, thiamine hydrochloride, sensory analysis, gnocchi dumplings

## Abstract

The aim of the study is to explore the possibility of vegetables being used as carriers of thiamine. The influence of carrier type (thiamine hydrochloride—TCh and thiamine pyrophosphate—TP) for the thiamine stability were investigated. Two varieties of pumpkin, Muscat and Hokkaido, as well as Cauliflower and Broccoli, were used as a matrix for the thiamine applied. The impregnated and freeze-dried vegetables were stored (230 days) with changing access to light (access to and restriction of light) and temperature (21 °C and 40 °C). The analyzed carriers were also used in the production of gnocchi dumplings. The content of thiamine was analyzed using the thiochromium method. In the study, consumer tests (*n* = 199) and sensory profiling were used to assess the impact of thiamine carriers on the sensory quality of gnocchi dumplings. It was found that the introduction of dried vegetables at the level of 30% allows for high sensory desirability of analyzed products, as well as suggesting the possibility of their frequent consumption. Such a product could potentially become an alternative to pork meat as a good source of thiamine. However, it should be noted that the thiamine losses may occur during the storage of dried vegetables and their culinary preparation.

## 1. Introduction

Thiamine (vitamin B1) is a water-soluble vitamin, occurring both as a free form, and bound as phosphate esters: thiamine pyrophosphate (TPP), thiamine monophosphate (TMP) and thiamine triphosphate (TDP) [1]. Thiamine deficiency leads to oxidative metabolism disorders, reduces the synthesis of adenosine triphosphate (ATP) and energy production, affecting sensitive organs, as well as altering their functions [1,2].

Thiamine plays a crucial role in energy metabolism. Moreover, this vitamin is also known for its catalytic activity in hexose-monophosphate pathways [3]. Other studies also indicate that thiamine therapy should be a crucial part of the treatment plan and health policies to prevent the development or progression of dementia and Alzheimer’s disease [4,5]. Very high doses of thiamine, i.e., up to 3 g/day are used in the treatment of numerous diseases [6,7,8,9]. According to Kunisawa, several neurological symptoms, occurring in other viral infections and neuroinflammatory states, may also be treated with thiamine [10].

Moderate amounts of thiamine can be found in most food products, albeit its most abundant sources constitute pork, legumes, whole grain products and nuts [11,12]. Small amounts of thiamine can also be synthesized by intestinal microorganisms [13].

Globally, many populations may be exposed to clinical or subclinical deficiencies of this vitamin. In the human body, thiamine reserves are depleted within 2 weeks after its food deficit, while clinical symptoms develop within the next three weeks [14,15]. Thiamine deficiency may result from its reduced absorption from the gastrointestinal tract, induced by illness or surgery, as well as increased metabolic demand [16]. [9,15,17,18]. Vitamin B1 deficits may be common both in the elderly (diseases, medication intake, absorption disorders) [15,19,20] and physically active young people—stress can induce biochemical thiamine deficiency [17,21,22]. Frequent deficiencies of this vitamin can be caused by its inadequate dietary intake. This may result from the intake of small portions of food, as well as the consumption of highly processed products, such as white flour or pasta produced from white flour. It is similar in the case of products such as tea, coffee, raw fish and crustaceans [9,17]. Certain nutritional studies suggest that vegetarian and vegan diets may have insufficient thiamine content [18]. This can arise due to an exclusive or dominant consumption of plant-based food—particularly highly processed products, as well as the absence of meat. It is worth noting that the consumption of animal products is decreasing, and the popularity of the vegan diet is growing worldwide. This trend is partially due to ecological aspects [19]. For a group of consumers who exclude or avoid certain categories of food of animal origin, alternative and attractive sources of many nutrients may be biofortificated vegetables and crops. However, this applies especially to fortification with mineral components [20,21,22]. It may also be possible that the introduction of increased levels of vitamins and minerals into these pathways could have detrimental consequences to the plant [23]. Furthermore, the most frequent food vehicle for fortification with thiamine is wheat flour, followed by maize flour and rice [17,24]. Fortification of vegetables may also constitute an attractive alternative source of many nutrients, including thiamine, for all groups of consumers. The selection of plant-based products also allows for an increase in the dietary intake of fiber, of which dieticians recommend raising daily amounts in an adult’s diet to 25–35 g [25]. Vegetables such as pumpkin, broccoli and cauliflower are an important natural source of dietary fiber [26,27]. These vegetables also contain polyphenolic compounds [28]. Numerous in vivo and in vitro studies have demonstrated that isothiocyanates and indole derivatives exhibit chemopreventive activity against cancer [27,28].

The enrichment of pasta with vegetables could be considered a strategy to increase vegetable intake without significantly changing the eating habits of the population [26].

Products like pasta and noodles are very popular foods in many countries all over the world [29,30].

However, these products predominantly contain starch. Therefore, in many studies attempts have been made to improve the nutritional properties of these products. 

The consumption of pasta enriched with thiamine impregnated vegetables could be used to create potentially functional food that may help to reduce thiamine deficiencies and some diseases. 

The aim of the study is to explore the possibility of employing two varieties of pumpkin, and cauliflower and broccoli as carriers of thiamine. The influence of the types of the vitamin (thiamine hydrochloride and thiamine pyrophosphate) for the thiamine stability, were investigated.

## 2. Materials and Methods

### 2.1. Material

Thiamine hydrochloride (TCh) and thiamine pyrophosphate (TP) constituted the sources of thiamine (Merc). Two varieties of pumpkin, Muscat and Hokkaido (*Cucurbita maxima Duch*.), as well as Cauliflower (*Brassica oleracea var. botrytis L*.) and Broccoli (*Brassica oleracea L*.), were used as a matrix for the thiamine applied. The vegetables contained thiamine in the range from 0.020 to 0.028 mg/100 g (PM 0.028; PH 0.021; B 0.020 C 0.022 mg/100 g). The products were purchased in the retail trade.

#### 2.1.1. Carriers Preparation and Rehydration

The vegetables were washed under running tap water. The pumpkin was peeled with stainless steel knives, and the seeds were removed. All the vegetables were cut into small pieces of approximately 4 × 4 × 4 cm. Next, the vegetables were steamed (100 °C; 5 min for broccoli and cauliflower, and 10 min for pumpkin) in a convection oven (Rational, Landsberg am Lech, Germany). The vegetables were subsequently drained and subjected to homogenization (homogenizer—Foss, Hilleroed, Denmark), to obtain a particle size of 250 μm. The impregnation process of vegetables consisted in their soaking (10 min) in a thiamine hydrochloride/thiamine pyrophosphate aqueous solution, at a ratio of 1:2 (*m*/*v*), using distilled water. Next, the samples were chilled for 60 min/4 °C. Then, the impregnated preparations were freeze-dried (approximately 48 h) to establish the moisture content at the level 4–5%. The dried vegetables were subjected to homogenization (homogenizer—Foss, Hilleroed, Denmark), to obtain a powder particle size of approximately 250 μm.

#### 2.1.2. Gnocchi Dumplings Formulations

The analyzed carriers were used in the production of gnocchi dumplings. Potatoes and flour were purchased in the retail trade. The potatoes were washed under running tap water. Next, the potatoes were cooked in a stainless steel pot using boiling tap water (1:2 (*m*/*v*)) for 25 min at 100 °C, until soft. Then the potatoes were drained and cooled at room temperature (20 ± 2 °C) for 15 min. The skin was removed and potatoes were mashed using a ricer (Westmark GmbH, Lennestadt-Elspe, Germany). Mashed potatoes were mixed with ingredients according to variants:WN—potato 80%; flour (10%); water (10%);MPTCh—potato 56%; flour (10%); water (10%); thiamine hydrochloride impregnated Muscat pumpkin (24%);MPTP—potato 56%; flour (10%); water (10%); thiamine pyrophosphate impregnated Muscat pumpkin (24%);HPTCh—potato 56%; flour (10%); water (10%); thiamine hydrochloride impregnated Hokkaido pumpkin (24%);HPTP—potato 56%; flour (10%); water (10%); thiamine pyrophosphate impregnated Hokkaido pumpkin (24%);CTCh—potato 56%; flour (10%); water (10%); thiamine hydrochloride impregnated cauliflower (24%);CTP—potato 56%; flour (10%); water (10%); thiamine pyrophosphate impregnated cauliflower (24%);BTCh—potato 56%; flour (10%); water (10%); thiamine hydrochloride impregnated broccoli (24%);BTP—potato 56%; flour (10%); water (10%); thiamine pyrophosphate impregnated broccoli (24%).

Preparations impregnated with thiamine were added to the batter in the hydrated form (at 1:3 ratios). After thorough mixing of the additives (approximately 10 min) pieces with a typical oval shape and similar weights (approximately 10 g) were formed.

Next, all the variants of gnocchi were divided, into two parts, for:Freezer storage—slow freezing until −21 °C (12 h; in a domestic freezer);Cooked—The gnocchi were steamed in a convection oven (Rational, Landsberg am Lech, Germany), with steam as a heating medium and using boiling tap water (initial water content—1:2 (*m*/*v*)) for 5 min at 100 °C. Then the gnocchi were drained and cooled at room temperature (20 ± 2 °C) for 10 min.

#### 2.1.3. Storage Conditions of Thiamine Sources

##### Preparations Impregnated

The impregnated and freeze-dried vegetables under investigation were stored in jars (black or clear glass, closed with screw top, d = 7 cm, h = 10 cm). The influence of storage conditions on the stability of TCh and TP was tested with the changing access of light and temperature (T): ▪ 21 AL—the access to light and temperature (21 ± 1 °C),▪ 21 RL—restriction of light and medium temperature (21 ± 1 °C),▪ 40 AL—the access to light and temperature (40 ± 1 °C),▪ 40 RL—restriction of light and temperature (40 ± 1 °C).

The thiamine contents in the investigated carriers were monitored on the selected storage days: 1, 30, 60, 120, 150, 180 and 230. 

##### Gnocchi Dumplings with Preparations Impregnated

The uncooked gnocchi samples were stored at −21 °C for 90 days in vacuum-sealed, medium density polyethylene bags. Storage time had the least effect on the sensory quality of this product (own unpublished data). In the gnocchi, the thiamine contents with carriers were monitored on the selected days of storage: 1, 15, 30, 45, 60, 75 and 90.

### 2.2. Methods

#### 2.2.1. Stability of Thiamine

On the set days of storage, samples were tested using the thiochromium method [31,32] which included the analysis of quantitative changes in the free (thiamine hydrochloride) and bound (thiamine pyrophosphate) form.

A Jenway model 6200 fluorometer (Jenway, Stone, UK) (input filter with maximum 365 nm and output filter with maximum 435 nm) was used for the measurement of thiochromium fluorescence. All determinations were made in duplicate.

#### 2.2.2. Sensory Analysis

Investigations were conducted in an appropriately designed and equipped laboratory of sensory analysis [33] at the Department of Gastronomy Science and Functional Foods, Poznan University of Life Sciences, Poland. The samples were coded with three-digit numbers, and the serving order of samples was random (program Analsens was used for coding and arrangement of serving order). Samples with an approximate mass of 30 g were placed in previously coded plastic containers (150 mL) and covered with lids. Unsweetened black tea (temp. ~45 °C) was used as a taste neutralizer between the samples.

The sensory profiling of taste was conducted by a 9-member trained panel. The 9 descriptors were adapted for taste (salty, vegetable, bitter, peppery, flour, potato, sweet, chemical and metallic). The intensity of each score was determined using a 10 cm linear scale with appropriate margin descriptions. For attributes, uniform margin denotations were applied: “undetectable—very intensive”. All samples were assessed in two independent replications.

Consumer traits were conducted on a group of 199 people aged 20–34. Women constituted 51% of the population analyzed. All subjects gave written informed consent to participate.

#### 2.2.3. Statistical Analysis

The results were analyzed statistically with the STATISTICATM PL 13.3 software (StatSoft, Tulusa, OK, USA).

The data were analyzed for statistically significant differences with the Tukey’s multiple range test (*p* ≤ 0.01). The thiamine contents were analyzed in 14 samples (two independent samples and seven measurements for each sample). The thiamine data were submitted to linear regression analysis and the goodness of fitting was evaluated on the basis of statistical parameters of fitting (R^2^ and a probability level of the models).

To predict the dynamics of changes in thiamine content in gnocchi and carriers during the storage, the half-life value (T_1/2_) was used. This is a term that describes the time within which the initial thiamine content decreased by half. The accuracy of the models was estimated using the coefficient of determination (R^2^) and root mean square error (RMSE). The significance level for all analysis was set at 5% [34,35].

For the overall evaluation of differences and similarities in sensory profiles of the tested samples, the analysis of main components (PCA—Principal Component Analysis) was used. The results were subjected to the one-way analysis of variance (Statistica Software v. 13, StatSoft, Tulusa, OK, USA) and Tukey’s test. Hypotheses were tested at α = 0.01.

## 3. Results

### 3.1. Changes in Content of Thiamine during Storage of Thiamine Sources

The results of stability changes in the thiamine (Figure 1, Table 1), showed that all vegetables analyzed constitute good carriers for both forms of thiamine. The analysis of the relation of thiamine content in carriers to the initial content in the vegetables showed the effectiveness at the level of 75–89%. The highest effectiveness of thiamine fortification was found for muscat pumpkin (89%). The effectiveness depended also on the forms of thiamine. The results showed lower content of thiamine pirophosphorane than hydrochloride compared to the initial content in the vegetables for hokkaido pumpkin (76%) and broccoli (77%). Figure 1 demonstrates thiamine content (mg of thiamine per kg of carrier) in selected forms (TCh and TP), stored for 6 months at 21 °C and 40 °C, with limited and unlimited access to light. After 6 months of storage, these products contained from 66 to 90% of thiamine. The highest thiamine content was found in samples of Hokkaido pumpkin, enriched with thiamine hydrochloride.

The analysis of all predictors’ influence on thiamine changes (half-life T (1/2; *p* < 0.05)), Table 1) revealed that the storage conditions, i.e., the temperature (T) and access to light (RH), significantly affected the loss of thiamine in all carriers. A statistical analysis of changes in thiamine content during storage proved that these alterations occurred in accordance with the first-order reaction kinetics (Table 1). It was found that the lower the temperature of storage and the greater the restriction of light, the lower the dynamics of thiamine loss.

The least significant loss of thiamine was observed at 21 °C with no access to light, regardless of the thiamine carrier and its form. The half-life of thiamine T (1/2) in such conditions was longer in the range of 2–9%, compared to 40 °C, especially for TP and Cauliflower.

The stability of the applied thiamine exhibited a statistically significant dependence (*p* < 0.05) also on its form (thiamine hydrochloride vs. thiamine pyrophosphate). The analysis of study results (*p* < 0.05) indicated that, regardless of the storage conditions and type of thiamine carrier, the lowest stability of thiamine was found for thiamine pyrophosphate. In the samples enriched with this form of thiamine, the analysis of the change dynamics showed that the rate of transformation increased by 4–7%.

### 3.2. Changes in Thiamine Content during Thermal Processing of Gnocchi Dumplings with Thiamine Sources

Thermal processing (steam cooking) of gnocchi, with the addition of all dried vegetables enriched with thiamine resulted in a statistically significant decrease (*p* < 0.001) of thiamine compared to the initial thiamine content before the cooking. The gnocchi before cooking contained thiamine in the range from 0.121 (MPTH) to 0.150 (CTP) mg/100 g. After cooking the changes in thiamine content were found in the range from 0.088 (HPTP) to 0.112 (WTH) mg/100 g. The covariance analysis (Figure 2) revealed no statistically significant effect (*p* < 0.05) on the type of thiamine carrier. The amounts of losses depended solely on the forms of thiamine (Figure 2). All samples exhibited higher losses of thiamine introduced in the form of TP. After thermal processing, the least significant loss of thiamine amounted to 25% and involved samples with the addition of dried pumpkin, enriched with PCH. Similar losses were determined for products with the addition of dried cauliflower (22%) and broccoli (22.6%), Muscat pumpkin (21.4%) and Hokkaido pumpkin (21.9%). The stability of thiamine applied on the dried pumpkins in the form of TP was 4% lower. Similar trends were found for dried cauliflower and broccoli.

### 3.3. Changes in Thiamine Content during Frozen Storage of Gnocchi Dumplings with Thiamine Sources

Storage of the analyzed gnocchi enriched with thiamine in freezing conditions resulted in a decrease of the thiamine content in the product (Figure 3, Table 2). However, it should be noted that the said losses were not significant. It was determined that all the dried vegetables analyzed may constitute good carriers for both forms of thiamine. After three months of storage, the loss of thiamine introduced into the product along with the dried vegetables amounted to 11–24%. The highest thiamine content was found in the samples of Hokkaido pumpkin, enriched with thiamine hydrochloride. A similar trend of heightened sensitivity involved thiamine applied in the form of TP. The analysis of change dynamics showed that the slowest rate (by 5%) of thiamine transformation occurred in the samples containing TCh.

### 3.4. The Effect of Adding the Vegetables Fortified with Thiamin to Gnocchi Dumplings on the Sensory Quality 

The designed gnocchi dumplings with thiamine carriers could be intended to supplement thiamine deficiencies in different groups of people. The results of previous studies suggest that the sensory properties of foods play a very important role in the selection of food and its amounts for consumption [36,37].

Consumer evaluation results (Table 3) showed that gnocchi dumplings both with the addition of and without thiamine carriers were characterized by similar and high desirability. The results of the analysis showed a similarity between evaluations of all product variables (Table 3, Figure 4). The desirability of the products was rated in the range between 6.8 and 7.5 points on the 10-point scale.

In the sensory profiling, perception of the following taste descriptors was defined and determined (salty, vegetable, bitter, peppery, flour, potato, sweet, chemical and metallic). The results of the sensory profiling of gnocchi dumplings with TCh and TP carriers and the control sample (without carriers) are presented in Table 3.

Principal component analysis (PCA) was used to study the relations between the taste attributes characteristic for sensory profiles of gnocchi dumplings (variables) and to derive factors according to which these variables can be classified. The PCA showed that the first factor (F1) was the most important element explaining variation in the data (98.9%). It is worth highlighting that F1 was strongly related to all taste attributes.

In the case of taste profiles (Figure 4 and Figure 5, and Table 3) it was found that the control samples and the samples with all thiamine carriers were characterized by the low intensity of salty, bitter, peppery, flour, sweet, chemical and metallic taste. The gnocchi dumplings without thiamine carriers were characterized by low intensity of vegetable taste but higher intensity of flour taste.

## 4. Discussion

It should be stated that thiamine losses in the dried vegetables analyzed were not high. Thiamine is one of the most labile vitamins [9,12,31,38,39,40]. The analysis of Japanese cooking meals have shown 50% losses of thiamine on average. In these studies it was found that the cooking losses of thiamine were particularly high in rice and green vegetables. [41]. Also, high losses of thiamine (45%) were observed in the samples of rice fortified with thiamine and cooked in a microwave oven [42]. It was also found that the content of thiamine was reduced by 50–70% for enriched polished and parboiled rice [43].

When analyzing the stability of thiamine during storage of dried vegetables, as well as products with their addition, it was found that thiamine in the form of pyrophosphate exhibits lower stability. This is confirmed by previous studies related to both high-temperature processing and storage [44,45]. This could be explained by the fact that a smaller amount of activation energy is required to break down the thiazole ring of thiamine [45,46,47]. The analysis of all predictors’ influence on thiamine changes during storage proved that these alterations occurred in accordance with the first-order reaction kinetics. Literature data also indicate the disintegration of the thiamine molecule according to first-order reactions [45]. The values of half-life thiamine periods indicate that the lower the temperature of storage, the lower the dynamics of thiamine loss. Earlier studies showed that most chemical compounds are increasingly active at higher temperatures [[9,48,49]]. Thiamine is one of the most labile vitamins. An increase in the temperature from 4 °C to 21 °C already accelerates the rate of its reactivity. Moreover, studies on iodine with thiamine, for example, confirmed its higher activity at higher temperatures and therefore, its higher losses [32,44].

Significant losses were found during the thermal processing of the finished product, with the addition of enriched, dried vegetables. The negative impact of cooking on the stability of thiamine must be associated with the degradative influence of high temperatures and the migration of water-soluble nutrients from the product to the environment. This phenomenon particularly favors cooking methods in which water constitutes a heating medium [32,50,51].

In this experiment, the application of thermal processing in a convection oven, with steam as a heating medium, retained higher amounts of thiamine in the gnocchi. The earlier study showed that the use of the convection oven resulted in lower losses of the thiamine content than traditional methods [38]. Thiamine is thermolabile and water-soluble therefore, according to Leskova et al. [52], methods that minimize direct contact of food with the cooking water have been found to be preferable to boiling for hydro-soluble vitamin retention.

It should be noted that the losses were approximate, or lower than in the case of products containing thiamine in its native form. During similar heating conditions of meat products, thiamine losses amounted to 50% [38,52].

On the sensory analysis it was found that all products with the addition of analyzed preparations were characterized by similar and high sensory desirability, independent of the variant of preparation added (*p* < 0.001). An earlier study has shown that bitter taste can decrease consumer acceptance of food products, particularly those for which this taste is not characteristic [43,53,54,55]. Metallic taste is also an important problem in food technology, for example with sweeteners. An earlier study showed that the problem with the incorporation of broccoli into food is that it can alter its sensory properties, especially bitter taste [37,56]. Our products with vegetables were characterized by a low intensity of bitter and metallic taste, which influenced the high consumer desirability of the products. An adverse effect on the sensory properties of these products has not been found. Consumer evaluation results of pasta with broccoli and pumpkin showed similar and high desirability [48,57,58]. The earlier research confirmed the high desirability of cereal flours in bakery products like cakes, cookies, bread, soups, sauces and instant noodles with the addition of pumpkin. Pumpkin flour is popular due to its highly-desirable flavor, sweetness and deep yellow-orange color [49,59]. The study of Drabińska confirms the successful addition of broccoli to mini sponge cakes [57].

It was found that the application of pumpkin increased the stability of thiamine during storage. It suggests the effect results from higher antioxidant activity of pumpkin. Earlier studies showed that the antioxidant activity of pumpkin products was significantly higher than for products with broccoli and cauliflower [50,51,60]. Further research on correlation between the antioxidant activity of the analyzed carriers and stability of thiamine is necessary to clarify this point.

The results may be interesting for nutritionists, as well as for food producers who offer food for the consumers at high risk of thiamine deficiency, for example people with Alzheimer’s disease, as well as vegans and vegetarians.

## 5. Conclusions

The analyzed possibility of employing all dried vegetables, enriched with thiamine to enrich food with thiamine appears to be effective. The highest effectiveness of fortification for muscat pumpkin has been found. The introduction of dried vegetables at the level of 30% allows for high sensory desirability of analyzed products, as well as suggesting the possibility of their frequent consumption. When analyzing the possibility of satisfying the thiamine demand by consuming the examined products, it should be noted that the thiamine losses may occur during the storage of dried vegetables and their culinary preparation. The consumption of 100 g of the analyzed product may constitute a source of thiamine at the level of 0.22–0.24 mg, which covers 25% of daily required intake. Such a product could potentially become an alternative to pork meat as a good source of thiamine.

## Figures and Tables

**Figure 1 foods-10-00578-f001:**
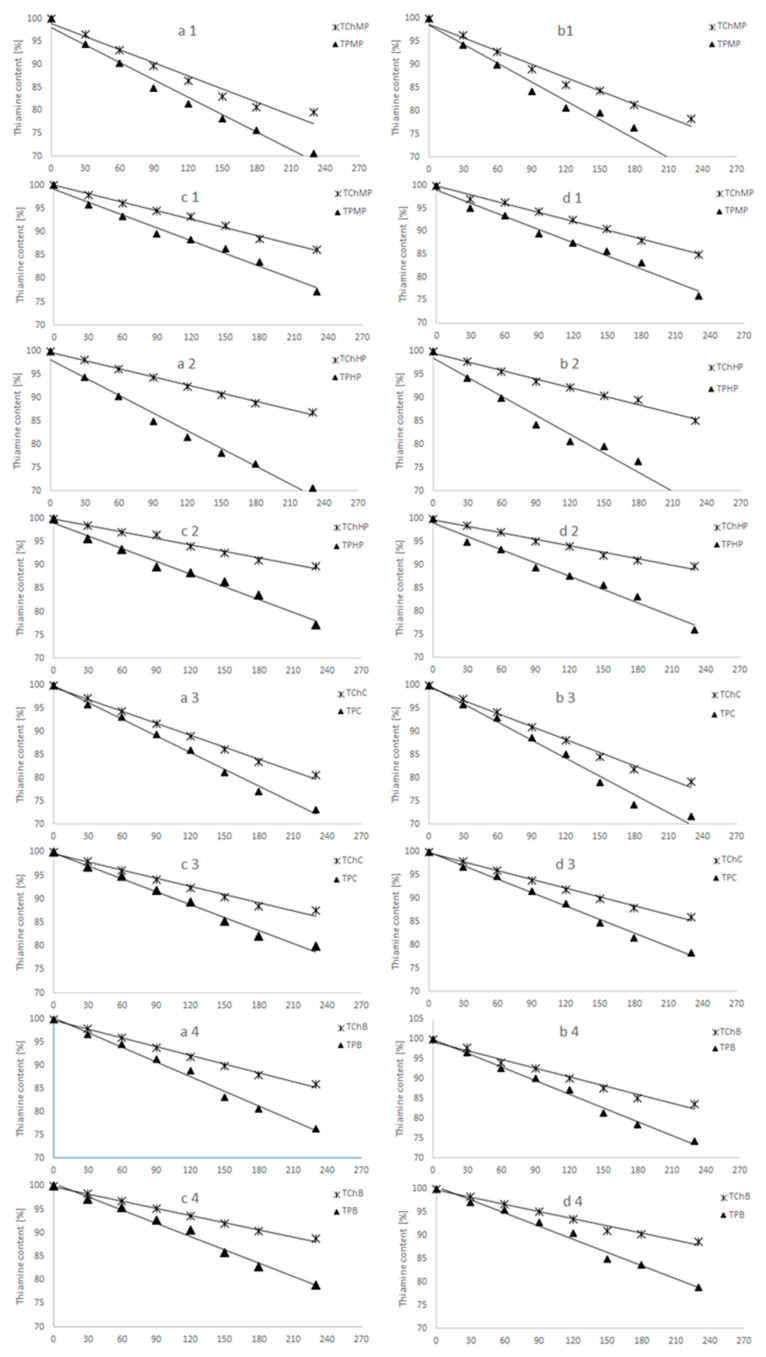
Thiamine content [%] in carriers: Muscat Pumpkin (1), Hokkaido Pumpkin (2), Cauliflower (3), Broccoli (4) of thiamine hydrochloride and thiamine pyrophosphate stored during 230 days in the presence of different storage conditions (a—40 °C and restriction of light; b—40 °C and access to light; c—21 °C and restriction of light; d—21 °C and access to light).

**Figure 2 foods-10-00578-f002:**
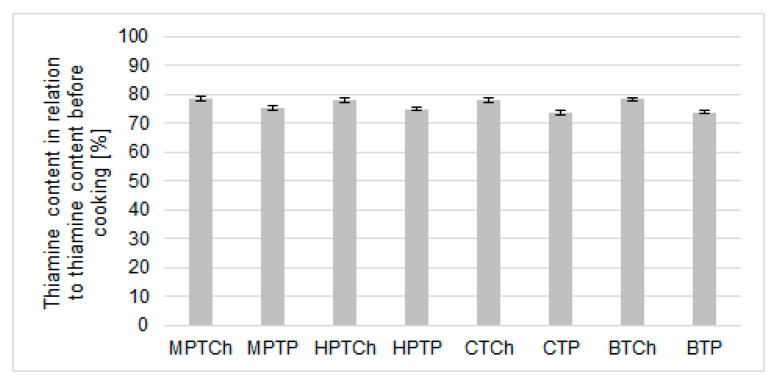
Thiamine content [%] after thermal processing (100 °C/5 min) of gnocchi dumplings with thiamine hydrochloride (TCh) and thiamine pyrophosphate (TP) sources (Muscat Pumpkin—MP, Hokkaido Pumpkin—HP, Cauliflower—C and Broccoli—B).

**Figure 3 foods-10-00578-f003:**
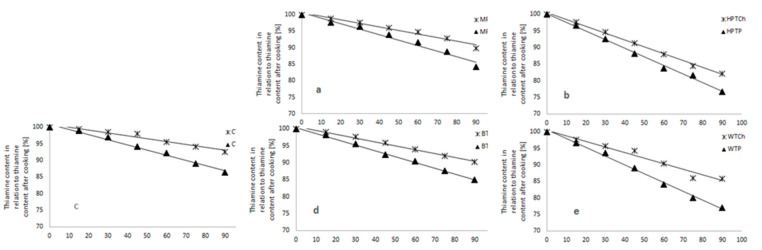
Thiamine content [%] in gnocchi dumplings with thiamine hydrochloride—TCh and thiamine pyrophosphate—TP carriers: (**a**)-Muscat Pumpkin (MP), (**b**)-Hokkaido Pumpkin (HP), (**c**)-Cauliflower, (**d**)-Broccoli and (**e**)-without carriers, during frozen storage. The 100% of thiamine content corresponds to the after-cooking thiamine content.

**Figure 4 foods-10-00578-f004:**
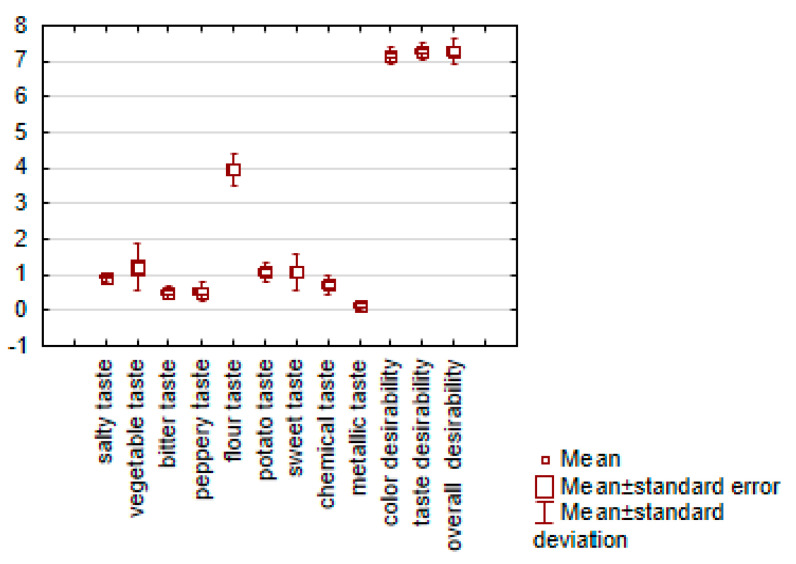
Box plot diagram of consumer desirability (color, taste and overall desirability) and attributes of taste profiles of gnocchi dumplings with thiamine hydrochloride and thiamine pyrophosphate carriers (Muscat Pumpkin, Hokkaido Pumpkin, Cauliflower, Broccoli) and the control sample (without carriers).

**Figure 5 foods-10-00578-f005:**
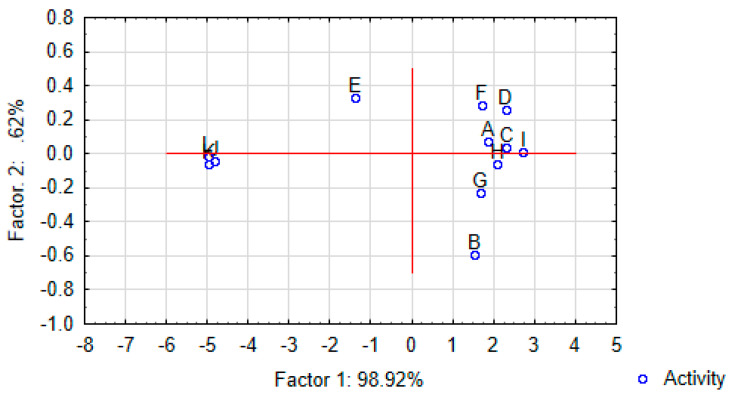
Map of the variants of gnocchi dumplings with thiamine hydrochloride and thiamine pyrophosphate carriers (Muscat Pumpkin, Hokkaido Pumpkin, Cauliflower, Broccoli) and the control sample (without carriers) into factors (“F1xF2”). Case-factor coordinates plots based on attributes of taste profiles and data from consumer analysis (A—salty taste; B—vegetable taste; C—bitter taste; D—peppery taste; E—flour taste; F—potato taste; G—sweet taste; H—chemical taste; I—metallic taste; J—taste desirability and K—overall desirability) (PCA analysis).

**Table 1 foods-10-00578-t001:** Dynamic of changes in thiamine content (mg TCh or TP kg^−1^) in carriers (Muscat Pumpkin, Hokkaido Pumpkin, Cauliflower and Broccoli) stored during 230-days in presence of different temperature and access to light conditions shown as values of half time of thiamine degradation (T_1/2_) and coefficients in regression equations.

Type of Vegetables			Storage Conditions	Values of the Dynamics of Changes in Thiamine Content during 230 Days				
				T_1/2_[days]	R*^2^*	RMSE	Y = ax + b	
							Coeff. a 24 h^−1^	b
**Muscat Pumpkin**	CCh	21 °C	RL	824.00	0.99	0.0050	−(6.3 ± 0.01) × 10^−4^	2.994 ± 0.0006
			AL	774.39	0.99	0.0047	−(6.8 ± 0.01) × 10^−4^	2.990 ± 0.0003
		40 °C	RL	500.94	0.97	0.0134	−(1.0 ± 0.02) × 10^−3^	2.959 ± 0.0004
			AL	501.19	0.98	0.0096	−(1.0 ± 0.01) × 10^−3^	2.955 ± 0.0004
	TP	21 °C	RL	527.10	0.98	0.0106	−(1.1 ± 0.04) × 10^−3^	3.419 ± 0.0002
			AL	504.35	0.98	0.0119	−(1.2 ± 0.09) × 10^−3^	3.414 ± 0.0005
		40 °C	RL	374.45	0.98	0.0429	−(1.6 ± 0.07) × 10^−3^	3.375 ± 0.0001
			AL	350.64	0.98	0.0186	−(1.7 ± 0.09) × 10^−3^	3.392 ± 0.0001
**Hokkaido Pumpkin**	TCh	21 °C	RL	1055.43	0.98	0.0076	−(5.3 ± 0.00) × 10^−4^	1.140 ± 0.0004
			AL	1047.13	0.99	0.0047	−(5.3 ± 0.00) × 10^−4^	1.137 ± 0.0003
		40 °C	RL	843.68	0.99	0.0052	−(6.7 ± 0.00) × 10^−4^	1.137 ± 0.0002
			AL	801.04	0.99	0.0072	−(7.0 ± 0.01) × 10^−4^	1.136 ± 0.0004
	TP	21 °C	RL	621.10	0.99	0.0352	−(8.6 ± 0.01) × 10^−4^	2.948 ± 0.0001
			AL	608.58	0.98	0.0095	−(8.6 ± 0.00) × 10^−4^	2.939 ± 0.0001
		40 °C	RL	582.09	0.97	0.0114	−(8.8 ± 0.00) × 10^−4^	2.916 ± 0.0001
			AL	536.65	0.97	0.0130	−(9.6 ± 0.00) × 10^−4^	2.924 ± 0.0003
**Cauliflower**	TCh	21 °C	RL	1055.43	0.98	0.0076	−(6.4 ± 0.01) × 10^−4^	3.002 ± 0.0004
			AL	1047.13	0.99	0.0047	−(7.0 ± 0.00) × 10^−4^	3.010 ± 0.0001
		40 °C	RL	843.68	0.99	0.0052	−(9.6 ± 0.00) × 10^−4^	3.005 ± 0.0001
			AL	801.04	0.99	0.0072	−(1.05 ± 0.01) × 10^−3^	3.006 ± 0.0001
	TP	21 °C	RL	539.15	0.99	0.0102	−(1.3 ± 0.06) × 10^−3^	3.965 ± 0.0001
			AL	511.50	0.99	0.0079	−(1.3 ± 0.04) × 10^−3^	3.978 ± 0.0004
		40 °C	RL	412.26	0.99	0.0365	−(1.7 ± 0.05) × 10^−3^	3.970 ± 0.0004
			AL	376.01	0.98	0.0168	−(1.8 ± 0.02) × 10^−3^	3.976 ± 0.0004
**Broccoli**	TCh	21 °C	RL	980.32	0.99	0.0836	−(5.5 ± 0.00) × 10^−4^	2.950 ± 0.0004
			AL	950.44	0.98	0.0051	−(5.6 ± 0.01) × 10^−4^	2.950 ± 0.0001
		40 °C	RL	782.43	0.99	0.0037	−(6.9 ± 0.00) × 10^−4^	2.948 ± 0.0001
			AL	652.84	0.98	0.0084	−(8.1 ± 0.01) × 10^−4^	2.936 ± 0.0000
	TP	21 °C	RL	532.03	0.99	0.0393	−(1.1 ± 0.02) × 10^−3^	3.127 ± 0.0003
			AL	530.41	0.99	0.0094	−(1.1 ± 0.05) × 10^−3^	3.126 ± 0.0000
		40 °C	RL	469.59	0.99	0.0094	−(1.2 ± 0.01) × 10^−3^	3.121 ± 0.0001
			AL	429.28	0.99	0.0082	−(1.3 ± 0.01) × 10^−3^	3.107 ± 0.0001

**Table 2 foods-10-00578-t002:** Dynamics of changes in thiamine content (mg TCh or TP kg^−1^) in gnocchi dumplings with TCh and TP carriers: Muscat Pumpkin, Hokkaido Pumpkin, Cauliflower, Broccoli during frozen storage shown as values of half time of thiamine degradation (T_1/2_) and coefficients in regression equations.

Type of Vegetables	Forms of Thiamine	Values of the Dynamics of Changes in Thiamine Content during 90 Days				
		T_1/2_[Days]	*R^2^*	*RMSE*	Y = ax + b	
					Coeff. a 24 h^−1^	b
**Muscat Pumpkin**	TCh	457.10	0.97	0.0056	−(1.0 ± 0.00) × 10^−4^	1.100 ± 0.0002
	TP	296.76	0.98	0.0009	−(1.8 ± 0.01) × 10^−4^	1.116 ± 0.0000
**Hxkkaidx Pumpkin**	TCh	243.90	0.95	0.0196	−(8.0 ± 0.01) × 10^−5^	1.096 ± 0.0000
	TP	192.06	0.99	0.0006	−(1.7 ± 0.01) × 10^−4^	1.117 ± 0.0004
**Cauliflower**	TCh	560.49	1.00	0.0003	−(2.0 ± 0.00) × 10^−4^	1.105 ± 0.0005
	TP	324.08	1.00	0.0005	−(2.3 ± 0.00) × 10^−4^	1.092 ± 0.0005
**Broccoli**	TCh	446.99	0.99	0.0003	−(1.0 ± 0.01) × 10^−4^	1.094 ± 0.0007
	TP	294.44	0.99	0.0002	−(1.6 ± 0.01) × 10^−4^	1.097 ± 0.0007

**Table 3 foods-10-00578-t003:** Mean scores (*n* = 14) of sensory taste profiling and desirability (*n* = 199) of gnocchi dumplings with vegetables fortified with thiamine hydrochloride (TCh) and thiamine pyrophosphate (TP) and the control sample (without vegetable fortified of thiamine addition).

	WTCh	WTP	MPTCh	MPTP	HPTCh	HPTP	CTCh	CTP	BTCh	BTP
salty taste	1.00 ^a^ * ± 0.22	1.00 ^a^ ± 0.21	1.00 ^a^ ± 0.23	0.90 ^a^ ± 0.25	0.90 ^a^ ± 0.30	1.00 ^a^ ± 0.26	0.90 ^a^ ± 0.20	1.00 ^a^ ± 0.17	0.70 ^a^ ± 0.20	0.90 ^a^ ± 0.15
vegetable taste	0.00 ^a^ * ± 0.0	0.00 ^a^ ± 0.0	1.40 ^b^ ± 0.25	1.60 ^b^ ± 0.30	1.70 ^a^ ± 0.40	1.70 ^a^ ± 0.32	1.50 ^a^ ± 0.15	1.50 ^a^ ± 0.23	1.40 ^a^ ± 0.42	1.40 ^a^ ± 0.35
bitter taste	0.40 ^a^ * ± 0.22	0.40 ^a^ ± 0.14	0.50 ^a^ ± 0.20	0.40 ^a^ ± 0.21	0.40 ^a^ ± 0.15	0.40 ^a^ ± 0.11	0.40 ^a^ ± 0.22	0.50 ^a^ ± 0.22	0.90 ^a^ ± 0.25	0.80 ^a^ ± 0.25
peppery taste	0.90 ^a^ ± 0.21	1.00 ^a^ ± 0.22	0.60 ^a^ ± 0.20	0.80 ^a^ ± 0.20	0.40 ^a^ ± 0.16	0.40 ^a^ ± 0.15	0.30 ^a^ ± 0.12	0.30 ^a^ ± 0.14	0.30 ^a^ ± 0.10	0.30 ^a^ ± 0.15
flour taste	4.60 ^a^ ± 0.32	4.80 ^a^ ± 0.31	4.00 ^a^ ± 0.48	4.00 ^a^ ± 0.40	3.90 ^a^ ± 0.85	3.60 ^a^ ± 0.64	3.50 ^a^ ± 0.45	3.50 ^a^ ± 0.54	3.70 ^a^ ± 0.50	4.00 ^a^ ± 0.62
potato taste	1.60 ^b^ ± 0.22	1.50 ^b^ ± 0.31	0.80 ^a^ ± 0.22	0.80 ^a^ ± 0.24	0.90 ^a^ ± 0.25	0.90 ^a^ ± 0.24	1.00 ^b,a^ ± 0.22	1.20 ^b^ ± 0.21	1.10 ^b,a^ ± 0.22	1.10 b,^a^ ±0.20
sweet taste	0.70 ^a^ ± 0.31	0.60 ^a^ ± 0.25	2.00 ^a^ ± 0.54	1.90 ^a^ ± 0.25	1.10 ^a^ ± 0.53	1.20 ^a^ ± 0.25	1.00 ^a^ ± 0.45	1.00 ^a^ ± 0.54	0.80 ^a^ ± 0.43	0.60 ^a^ ± 0.54
chemical taste	0.40 ^a^ ± 0.22	0.50 ^a^ ± 0.25	0.60 ^a^ ± 0.32	0.60 ^a^ ± 0.22	0.50 ^a^ ± 0.15	0.60 ^a^ ± 0.6	1.10 ^a^ ± 0.40	0.90 ^a^ ± 0.43	0.90 ^a^ ± 0.35	1.10 ^a^ ± 0.41
metallic taste	0.00 ^a^ ± 0.00	0.00 ^a^ ± 0.00	0.20 ^a^ ± 0.13	0.20 ^a^ ± 0.12	0.00 ^a^ ± 0.00	0.00 ^a^ ± 0.00	0.40 ^a^ ± 0.20	0.20 ^a^ ± 0.14	0.20 ^a^ ± 0.10	0.20 ^a^ ± 0.11
taste desirability	7.49 ^a^ ± 0.95	7.47 ^a^ ± 0.96	6.96 ^a^ ± 0.67	6.99 ^a^ ± 1.14	6.95 ^a^ ± 1.00	7.06 ^a^ ± 0.96	7.54 ^a^ ± 0.80	7.54 ^a^ ± 0.78	7.44 ^a^ ± 0.56	7.41 ^a^ ± 0.98
overall desirability	7.58 ^a^ ± 0.95	7.51 ^a^ ± 0.96	6.82 ^a^ ± 0.70	6.86 ^a^ ± 0.99	6.97 ^a^ ± 0.97	6.97 ^a^ ± 0.87	7.60 ^a^ ± 0.85	7.60 ^a^ ± 0.98	7.53 ^a^ ± 0.78	7.44 ^a^ ± 0.83

* Different letters denote a significant difference for means (*n* = 14), at a α ≤ 0.05.

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
