# Peer review of "Pumpkin, Cauliflower and Broccoli as New Carriers of Thiamine Compounds for Food Fortification"

_foods, 2021, doi:10.3390/foods10030578_

Round 1

Reviewer 1 Report

The research intitled “Pumpkin and other vegetables as new carriers of thiamine compounds for food fortification” shows quite interesting results about the possibility of employing several varieties of pumpkin, cauliflower and broccoli as carriers of thiamine. Some major remarks need to be addressed:

  • The introduction is too long and sometimes repetitive. I recommend shortening it a lot because it is even longer than the discussion.
  • I recommend including “cauliflower” and “broccoli” in the title.
  • It is not very clear why we should add thiamin-fortified vegetables to gnocchi and not directly fortify gnocchi dumplings with thiamin. Following this reflection, I recommend that the authors better explain the purpose and conclusions of the study.
  • Did the authors quantify thiamnine in pumpkin, broccoli, and cauliflowers? It seems to me an important fact
  • Lines 41-42: The two articles cited did not demonstrated that the thiamine may facilitate an appropriate immune response during SARS-CoV-2 infection.

In the article “The Role of Vitamins on the Prevention and/or Treatment 419 of COVID-19 Infection; a Systematic” the authors found that the percentage of studies about association with COVID-19 was 0%.

In the article “Therapeutic Prospects for Th-17 Cell Immune Storm Syndrome and Neurological Symptoms in COVID-19: Thiamine Efficacy and Safety, In-vitro Evidence and Pharmacokinetic Profile” the authors investigated the effect of a three-week 200 mg dose of thiamine in lowering the Th17 response in patients with inflammation due to heavy alcohol drinking and they speculated that thiamine could be repurposed for treating the cytokine/immune storm of COVID-19, but they concluded that further studies using thiamine as an interventional/prevention strategy in severe COVID-19 patients are needed.

Lines 43-46: In the article "Mode of Bioenergetic Metabolism during B Cell Differentiation in the Intestine Determines the Distinct Requirement for Vitamin B1" there is no mention of SARS-CoV-2 infection.

Author Response

Poznań, 22th February, 2021

Dear Reviewer,

Reference: ID:  foods-1102668 entitled "Pumpkin and other vegetables as new carriers of thiamine compounds for food fortification”

We would like to thank you for your comprehensive review of our paper. The answer was given to all the comments.

The answers to reviewer comments and suggestions to the author's amendments were introduced to work, as suggested.

We hope that after these corrections you will find our paper suitable to publish in Foods Journal.

KRYSTYNA SZYMANDERA-BUSZKA

The corresponding author

         Question:

The introduction is too long and sometimes repetitive. I recommend shortening it a lot because it is even longer than the discussion.

Answer:   

  1. The sentence “It was found that appetite loss is associated with vitamin B1 deficiency in elderly Japanese patients living in rural areas.” has been deleted
  2. The sentence “Vitamin B1 deficiency commonly occurs in diabetics, pregnant and lactating women, smokers, alcoholics, as well as people on a high carbohydrate diet” has been deleted.
  3. The sentence “This trend is partially due to ecological aspects, because meat production is considered to be responsible for unsustainable use of resources and pollution, which is result of the inefficient conversion of proteins of plant origin to proteins of animal origin” has been corrected to “This trend is partially due to ecological aspects”
  4. In the sentence “For a group of consumers who exclude certain categories of food of animal origin as well as restrict or avoid meat and its products, alternative and attractive source of many nutrients may be biofortificated vegetables and crops.” has been deleted “as well as restrict or avoid meat and its products”.

The new sentence “For a group of consumers who exclude or avoid certain categories of food of animal origin, alternative and attractive source of many nutrients may be biofortificated vegetables and crops.”

  1. The sentences “This applies especially to fortification with mineral components [28–30]. However, it may be possible that the introduction of increased levels of vitamins and minerals into these pathways could have detrimental consequences to the plant. “   have been corrected to ” However, this applies especially to fortification with mineral components. It may also be possible that the introduction of increased levels of vitamins and minerals into these pathways could have detrimental consequences to the plant.”
  2. The sentences “The most frequent food vehicle for fortification with thiamine is wheat flour, followed by maize flour and rice. The selection of the plant-based product allows for an increase in the dietary intake of fibre.” have been corrected to “Also, the most frequent food vehicle for fortification with thiamine is wheat flour, followed by maize flour and rice. The selection of the plant-based product also allows for an increase in the dietary intake of fibre.”

Question: I recommend including “cauliflower” and “broccoli” in the title.

Answer:  The title has been corrected to: “Pumpkin, cauliflower and broccoli as new carriers of thiamine compounds for food fortification”.  Also, has been corrected the sentence of the aim: “The aim of the study was the possibility of employing several varieties of pumpkin and other vegetables as carriers of thiamine.” to “The aim of the study was the possibility of employing two varieties of pumpkin,  and cauliflower and broccoli as carriers of thiamine.”

Question: It is not very clear why we should add thiamin-fortified vegetables to gnocchi and not directly fortify gnocchi dumplings with thiamin. Following this reflection, I recommend that the authors better explain the purpose and conclusions of the study.

Answer: In the Introductions, after sentences “Fortification of vegetables may also constitute an attractive alternative source of many nutrients, including thiamine, for all groups of consumers. The selection of the plant-based product also allows for an increase in the dietary intake of fiber. Dieticians recommend raising its daily amount in an adult's diet to 25-35 g” have been added  „Such vegetables as pumpkin, broccoli and cauliflower are an important natural source of dietary fiber. These vegetables also contain of polyphenolic compounds. Numerous in vivo and in vitro studies have demonstrated that isothiocyanates and indole derivatives exhibit chemopreventive activity against cancer. Pumpkin flour is popular due to its highly-desirable flavour, sweetness and deep yellow-orange color. The enrichment of pasta with vegetable could be considered a strategy to increase vegetable intake without significantly changing the eating habits of the population.  The products type of pasta and noodles are very popular foods in many countries all over the world. However,  these products may contain predominantly starch. Therefore in many studies are attempted to improve the nutritional properties of these products. The consumption of pasta with thiamine impregnated vegetables enriched  could be used to create potentially functional food that may help to reduce thiamine deficiencies and some diseases.”

Question: Did the authors quantify thiamine in pumpkin, broccoli, and cauliflowers? It seems to me an important fact

Answer: Yes, the content of thiamine in vegetables was determined.

Content of thiamine:

pumpkin Muscat -  0.028mg/100g

pumpkin Hokkaido - 0.021mg/100g

broccoli - 0.020mg/100g;

cauliflowers - 0.022mg/100g

In the Material section has been written:

“The vegetables contained thiamine in the range from 0.020 to 0.028 mg/100g (PM 0.028; PH 0.021; B 0.020 C 0.022 mg/100g)”

Question:

Lines 41-42: The two articles cited did not demonstrated that the thiamine may facilitate an appropriate immune response during SARS-CoV-2 infection.

In the article “The Role of Vitamins on the Prevention and/or Treatment 419 of COVID-19 Infection; a Systematic” the authors found that the percentage of studies about association with COVID-19 was 0%.

In the article “Therapeutic Prospects for Th-17 Cell Immune Storm Syndrome and Neurological Symptoms in COVID-19: Thiamine Efficacy and Safety, In-vitro Evidence and Pharmacokinetic Profile” the authors investigated the effect of a three-week 200 mg dose of thiamine in lowering the Th17 response in patients with inflammation due to heavy alcohol drinking and they speculated that thiamine could be repurposed for treating the cytokine/immune storm of COVID-19, but they concluded that further studies using thiamine as an interventional/prevention strategy in severe COVID-19 patients are needed.

Answer: The sentence  “Moreover, literature data indicate that a suitable level of thiamine may facilitate an appropriate immune response during SARS-CoV-2 infection” has been corrected to “Moreover, literature data suggest that a suitable level of thiamine may facilitate an appropriate immune response during SARS-CoV-2 infection”  have been deleted

Question:

Lines 43-46: In the article "Mode of Bioenergetic Metabolism during B Cell Differentiation in the Intestine Determines the Distinct Requirement for Vitamin B1" there is no mention of SARS-CoV-2 infection.

Answer: It is true that, in this publication, the authors speculated about thiamine and COVID-19 and therefore the sentence: "Thiamine deficiency may result in insufficient antibody response, followed by more severe symptoms during SARS-CoV-2 infection"

The sentence has been deleted.

Reviewer 2 Report

The objective of the study is to investigate the incorporation potential of thiamine in vegetables, with the aim to develop plant-based functional foods. The influence of carrier type on thiamine stability is also evaluated. Pumpkin, cauliflower and broccoli were selected as the case studies. The study is interesting and well designed. However, there are several issues that the authors should address.

The authors should elaborate more on the Discussion section, including comparisons with relevant studies from the literature.

Additionally, the selection of the tested food products should be justified. Statistical analysis should be used in order to indicate the statistically significant differences between the tested parameters and carriers.

Please add statistics (standard deviations) in Figure 1 and Figure 3 data points.

Author Response

Poznań, 22th February, 2021

Dear Reviewer,

Reference: ID:  foods-1102668 entitled "Pumpkin and other vegetables as new carriers of thiamine compounds for food fortification”

We would like to thank you for your comprehensive review of our paper. The answer was given to all the comments.

The Title has been corrected to “Pumpkin, cauliflower and broccoli as new carriers of thiamine compounds for food fortification” - according to suggestions by another Reviewer.

The answers to reviewer comments and suggestions to the author's amendments were introduced to work, as suggested.

We hope that after these corrections you will find our paper suitable to publish in Foods Journal.

KRYSTYNA SZYMANDERA-BUSZKA

The corresponding author

         Question: The authors should elaborate more on the Discussion section, including comparisons with relevant studies from the literature.

Answer: The sentences have been added:

“The earlier study indicates that the activation energy, in rosehip nectar, for thiamine is lower than  L-ascorbic acid and riboflavin. It was found that between 70 to 95 0C were found to be 36.38, 55.30, and 37.15 kJ/mol, respectively.” “The analysis of Japanese meals have shown losses of thiamin after cooking in average 50%. In these studies was found that the cooking losses of thiamin were particularly large in rice and green vegetables.  Also, high losses of thiamine (45%) were observed in the samples of fortified with thiamine rice and cooking in a microwave oven. It was also found that he contente of thiamine was reduced by 50-70% for enriched polished and parboiled rice. “

“On the sensory analysis it was found that all products with the addition of analyzed preparations were characterized by similar and high sensory desirability, independently of the variant of preparation added (p< 0.01). The earlier study has shown that the bitter taste can decrease consumer acceptance of food products, particularly those for which this taste is not characteristic. Also, metallic taste is an important problem in food technology, for example with sweeteners. The earlier study showed the problem with the incorporation of broccoli into food is that it can alter its sensory properties, especially bitter taste. Our products with vegetables were characterized by a low intensity of bitter and metallic taste, which influenced the high consumer desirability of the products. It is not found an adverse effect on the sensory properties of these products.  Also, consumer evaluation results of pasta with broccoli and pumpkin showed similar and high desirability. Bhat and Bhat, (2013) confirmed the high desirability of cereal flours in bakery products like cakes, cookies, bread, soups, sauces, instant noodle with the addition of pumpkin. The study of Grabowska also confirms the successful addition of broccoli to mini sponge cakes”

Question: Additionally, the selection of the tested food products should be justified.

Answer:  In the Introduction sections, after sentences “Fortification of vegetables may also constitute an attractive alternative source of many nutrients, including thiamine, for all groups of consumers. The selection of the plant-based product also allows for an increase in the dietary intake of fiber. Dieticians recommend raising its daily amount in an adult's diet to 25-35 g” have been added  „Such vegetables as pumpkin, broccoli and cauliflower are an important natural source of dietary fiber. These vegetables also contain of polyphenolic compounds. Numerous in vivo and in vitro studies have demonstrated that isothiocyanates and indole derivatives exhibit chemopreventive activity against cancer. Pumpkin flour is popular due to its highly-desirable flavour, sweetness and deep yellow-orange color. The enrichment of pasta with vegetable could be considered a strategy to increase vegetable intake without significantly changing the eating habits of the population. The products type of pasta and noodles are very popular foods in many countries all over the world. However, these products may contain predominantly starch. Therefore in many studies are attempted to improve the nutritional properties of these products. The consumption of pasta with thiamine impregnated vegetables enriched  could be used to create potentially functional food that may help to reduce thiamine deficiencies and some diseases.”

Considering the portion size and the percentage of vegetables that can be added, vegetable pasta can significantly contribute to the recommended vegetable intake per day. The incorporation of fruits and vegetables into regularly eaten products is a food design strategy that leads to several advantages. Pasta is a staple food eaten daily or weekly that constitutes a dominant moiety of the diet in many countries.

Question: Statistical analysis should be used in order to indicate the statistically significant differences between the tested parameters and carriers.

Answer: Tables 1-2 and figure 2 have been corrected.  The probability level of the models has been written. In the text has been written values of statistical analysis (p ≤ 0.01 or p< 0.01). “…independently of the variant of preparation added (p< 0.01)”.

Reviewer 3 Report

Comments and Suggestions for Authors

The manuscript entitled "Pumpkin and other vegetables as new carriers of thiamine compounds for food fortification" is not worth right now due to its messy organization, incompletely experimental part, and weakly topic of results.

Specific comments

Line: 17: The storage temperature of 45ºC does not correspond to the temperature described in the methodology and results (40ºC), which one is correct?

Line 83: The carriers are the vegetables or the type of thiamine? I suggest changing “the influence of carrier type…” to “the influence of the vitamins type…”

The “Materials and Methods” part needs to be improved. The methodology requires rewording and other writing in some places. The explanation it is messy and needs to be presented in different sections to clarify the whole procedure. For example: Carriers preparation and rehydration; Gnocchi dumplings formulations; Storage conditions; Sensory analysis; Statistical analysis.

Explain accurately the different step procedures (cleaning, peeling, cooking) detailing the operational conditions (time, temperature ...), the cooking conditions, equipment used in each step, water used (distilled, bidistilled …), among others. The homogenization part needs a deeper explanation as to which equipment was used, the conditions, the final product obtained …

Line 119: Explain how you cooked and frozen the gnocchi: conditions, equipment, temperatures, time …

Line 122: Describe the storage conditions for both types of carriers (vegetables and gnocchi) as refrigerated conditions, explaining why the storage days are different for each carrier …

Line 135: Describe what you were determined in this study, describe briefly the determination, and specify which means the “free” and “bound” forms.

Line 164: Explain better the purpose of using the T1/2.

Lines 165 to 168: The sentence is not needed: “The half-life was calculated from simple linear regression equation y = ax + b; where: y – dependent variable, x - independent variable, a - independent variable coeff./slope of the line, b intercept.”,

The “Results” section needs coherency and improvement. Table and Figure content need to be clarified.

Table 1: Add the variation interval (±), add the slop (a) and intercept (b) significant figures.

Figure 1: The experimental points do not be connected, add the fitted line and the abscise units. The graphic need to be improved in size, and the colors used made reading difficult.

Line 240: Explain the thiamine initial content, before and after, the cooking. Clarify if the thiamine evolution in Figure 2 is calculated regarding the initial thiamine content before the cooking or if the initial thiamine content after the cooking.

Line 243: Describe the steam cooking procedure in the “material and methods” section.

Figure 2: The 100% of thiamine content corresponds to the “before-cooking” or “after-cooking” thiamine content? Because the results from Figure 2 does not correspond to the results shown in Figure 3.

Line 260: Describe the freezing procedure in the “material and methods” section.

Line 264: Why are not significant the losses?? Almost 25% of the initial thiamine content disappear (according to Figure 3).

Line 267: The results presented not match with the T1/2, which is the focus of the development of the results.

Figure 3: The experimental points do not be connected, add the fitted line and the abscise units. The graphic need to be improved in size, and the colors used made reading difficult. The results did not match with the results from Figure 2.

Table 2: Add the variation interval (±), add the slop (a) and intercept (b) significant figures. Why the T1/2 were extrapolated? (The storage days were not enough to find the real T1/2?), and the extrapolation was not the best option.

Table 3. Consider the significant figures according to the variation interval.

Line 359: The cited experiment is yours or from the cited authors (48-50)?

Line 360: Describe in MM the “convection oven” used in this study.

Line: 371-372: In this study, the antioxidant activity was not evaluated, therefore, it is impossible to find a “good” correlation between antioxidant activity and thiamine stability.

Author Response

Poznań, 22th February, 2021

Dear Reviewer,

Reference: ID:  foods-1102668 entitled "Pumpkin and other vegetables as new carriers of thiamine compounds for food fortification”

We would like to thank you for your comprehensive review of our paper. The answer was given to all the comments.

The Title has been corrected to “Pumpkin, cauliflower and broccoli as new carriers of thiamine compounds for food fortification” - according to suggestions by another Reviewer.

The answers to reviewer comments and suggestions to the author's amendments were introduced to work, as suggested.

We hope that after these corrections you will find our paper suitable to publish in Foods Journal.

KRYSTYNA SZYMANDERA-BUSZKA

The corresponding author

         The manuscript entitled "Pumpkin and other vegetables as new carriers of thiamine compounds for food fortification" is not worth right now due to its messy organization, incompletely experimental part, and weakly topic of results.

Specific comments:

Question:

Line: 17: The storage temperature of 45ºC does not correspond to the temperature described in the methodology and results (40ºC), which one is correct?

Answer: The error has been corrected (45ºC to  40ºC).

Question:

Line 83: The carriers are the vegetables or the type of thiamine? I suggest changing “the influence of carrier type…” to “the influence of the vitamins type…”  

Answer: The sentence “The influence of carrier type (thiamine hydrochloride and thiamine pyrophosphate) for the thiamine stability, were investigated.” has been corrected to   “The influence of the types of the vitamin (thiamine hydrochloride and thiamine pyrophosphate) for the thiamine stability, were investigated.”

Question:

The “Materials and Methods” part needs to be improved. The methodology requires rewording and other writing in some places. The explanation it is messy and needs to be presented in different sections to clarify the whole procedure. For example: Carriers preparation and rehydration; Gnocchi dumplings formulations; Storage conditions; Sensory analysis; Statistical analysis.

Answer: The 2nd chapter "Methods and Material" has been divided into parts:

2.1. Material

2.1.1. Carriers preparation and rehydration

2.1.2. Gnocchi dumplings formulations

2.1.3. Storage conditions of thiamine sources

2.1.3.1.Preparations impregnated

2.1.3.2.Gnocchi dumplings with preparations impregnated

2.2. Methods

2.2.1. Stability of thiamine

2.2.2. Sensory analysis

2.2.3. Statistical analysis

In “Material” section has been added: “Thiamine hydrochloride (TCh) and thiamine pyrophosphate (TP) constituted the sources of thiamine (Merc). Two varieties of pumpkin Muscat and Hokkaido (Cucurbita maxima Duch.), as well as Cauliflower (Brassica oleracea var. botrytis L.) and Broccoli (Brassica oleracea L.), were used as a matrix for the thiamine applied. The vegetables contained thiamine in the range from 0.020 to 0.028 mg/100g (PM 0.028; PH 0.021; B 0.020 C 0.022 mg/100g). The products had been purchased in the retail trade.”

Question:

Explain accurately the different step procedures (cleaning, peeling, cooking) detailing the operational conditions (time, temperature ...), the cooking conditions, equipment used in each step, water used (distilled, bidistilled …), among others. The homogenization part needs a deeper explanation as to which equipment was used, the conditions, the final product obtained …

Answer:  The step procedures has been corrected.  The sentences “The cleaned, peeled and cooked vegetables were subsequently drained and subjected to homogenization. The impregnation process of vegetables consisted in their soaking in a thiamine hydrochloride/thiamine pyrophosphate aqueous solution at a ratio of 1:2 (m/v) and was storage among 60min/40C. Then the impregnated preparations were freeze-dried (48h).” have been corrected to “The vegetables were washed under running tap water. The pumpkin was peeled with stainless steel knives, and the seeds were removed. The all vegetables were cut into small pieces of approximately  4×4×4 cm. Next, the vegetables were steamed (100oC;  5 min for broccoli and cauliflower, and 10 min for pumpkin) in a convection oven (Rational, Landsberg am Lech, Germany). The vegetables were subsequently drained and subjected to homogenization (homogenizer - Foss, Hilleroed, Denmark), to obtain particle size of 250 μm. The impregnation process of vegetables consisted in their soaking in a thiamine hydrochloride/thiamine pyrophosphate aqueous solution, at a ratio of 1:2 (m/v), using distilled water.  Next, the samples was storage among 60min/40C. Then the impregnated preparations were freeze-dried (approximately 48h) to establishing the moisture content at the level 4-5%. The dried vegetables were subjected to homogenization (homogenizer - Foss, Hilleroed, Denmark), to obtain powder particle size of approximately 250 μm.”

Question:

Line 119: Explain how you cooked and frozen the gnocchi: conditions, equipment, temperatures, time …

Answer:  The sentences “Potatoes were boiled for 25 min until soft. The skin was removed and potatoes were mashed using a ricer.” have been corrected to “Potatoes and flour were purchased in local marked. The potatoes were washed under running tap water. Next, the potatoes were cooked in a stainless steel pot using of boiling tap water (1:2 (m/v)) for 25 min at 1000C, until soft. Then the potatoes were drained and cooled at room temperature (20 ± 2 â—¦C) for 15 min. Then the skin was removed and potatoes were mashed using a ricer (Westmark GmbH; Lennestadt – Elspe; Germany).”

Also, the sentence „Then the gnocchi was cooked for 5 min at 1000C or stored frozen at −21°C (5 months) until further use.” has been corrected to „Next, all the variants of gnocchi were divided, into two parts, for:

Freezer storage - slow freezing until −21°C (12 hours; in a domestic freezer)

Cooked - The gnocchi was cooked in a stainless steel pot using of boiling unsalted tap water (1:3 (m/v)) for 5 min at 1000C. Then the gnocchi was drained and cooled at room temperature (20 ± 2 â—¦C) for 10 min.”

Question:

Line 122: Describe the storage conditions for both types of carriers (vegetables and gnocchi) as refrigerated conditions, explaining why the storage days are different for each carrier …

Answer:  the sentence „stored frozen at −21°C (3 months) until further use.” has been corrected to “The uncooked gnocchi samples were stored at −21°C for 90 days in vacuum-sealed, medium density polyethylene bags. Storage for the time had the least effect sensory quality of this product (own unpublished data). In the gnocchi, the thiamine contents with carriers were monitored on the selected days of storage: 1, 15, 30, 45, 60, 75, and 90. “

Question:

Line 135: Describe what you were determined in this study, describe briefly the determination, and specify which means the “free” and “bound” forms.

Answer:  As the "free thiamine" determined thiamine hydrochloride and as "bound" thiamine pyrophosphate.  Therefore, for the clarity the words "free thiamine" has been changed to "free" (thiamine hydrochloride) and "bound" to "bound (thiamine pyrophosphate)”. 

The new sentence: “On the set days of storage, the samples were determined using the thiochromium method [34,35] which included the analysis of quantitative changes in the free (thiamine hydrochloride) and bound (thiamine pyrophosphate) form.”

Line 164: Explain better the purpose of using the T1/2.

Answer:  The sentences has been corrected to:

“Reaction rate constants were determined by fitting the experimental data to first-order kinetic model and modelled as a function storage time. The thiamine data were submitted to linear regression analysis and the goodness of fitting was evaluated on the basis of statistical parameters of fitting (R2 and a probability level of the models). To predict the dynamics of changes in thiamine content in gnocchi and carriers during the storage the half-life value (T1/2) was used. It is a term that describes the time within which the initial thiamine content decreased by half. The accuracy of the models was estimated using the coefficient of determination (R2) and root mean square error (RMSE). The significance level for all analysis was set at 5%.”

Question:

Lines 165 to 168: The sentence is not needed: “The half-life was calculated from simple linear regression equation y = ax + b; where: y – dependent variable, x - independent variable, a - independent variable coeff./slope of the line, b intercept.”,

Answer:  The sentence has been deleted.

The “Results” section needs coherency and improvement. Table and Figure content need to be clarified.

Question:

Table 1: Add the variation interval (±), add the slop (a) and intercept (b) significant figures.

Answer:  The significant figures have been added, the values of T1/2  have been added

Question:

Figure 1: The experimental points do not be connected, add the fitted line and the abscise units. The graphic need to be improved in size, and the colors used made reading difficult.

Answer:  Figure 1 has been corrected

Question:

Line 240: Explain the thiamine initial content, before and after, the cooking. Clarify if the thiamine evolution in Figure 2 is calculated regarding the initial thiamine content before the cooking or if the initial thiamine content after the cooking.

Answer:  Before the sentence “The covariance analysis (Fig. 2) revealed no statistically significant effect (P <0.05) on the type of thiamine carrier.” has been added “The gnocchi before cooking contained thiamine in the range from 0.121 (MPTH) to 0.150 (CTP) mg/100g. After cooking the changes in thiamine content were found in the range from 0.088 (HPTP) to 0.112  (WTH) mg/100g.”

The content of thiamine in Figure 2 is calculated regarding the initial thiamine content before the cooking. Therefore, for the clarity the sentence “Thermal processing (cooking) of gnocchi, with the addition of all dried vegetables, enriched with thiamine, resulted in a statistically significant decrease (P <0.001) of thiamine.” has been corrected to “Thermal processing (cooking) of gnocchi, with the addition of all dried vegetables, enriched with thiamine, resulted in a statistically significant decrease (p <0.001) of thiamine, regarding the initial thiamine content before the cooking.”

To the Axis Title for Figure 2 has been written “Thiamine content in relation to thiamine content before cooking [%]”

Question:

Line 243: Describe the steam cooking procedure in the “material and methods” section.

Answer:  The “steam cooking” has been described in “Material and methods” section: “Cooked - The gnocchi was steamed in a convection oven in a convection oven (Rational, Landsberg am Lech, Germany), with steam as a heating medium and using boiling tap water (initial water content - 1:2 (m/v)) for 5 min at 1000C.”

Question:

Figure 2: The 100% of thiamine content corresponds to the “before-cooking” or “after-cooking” thiamine content? Because the results from Figure 2 does not correspond to the results shown in Figure 3.

Answer:  The content of thiamine in Figure 2 was calculated regarding the initial thiamine content before the cooking.  The content of thiamine in Figure 3 was calculated regarding the initial thiamine content after the cooking. The 100% of thiamine content corresponds to the after-cooking thiamine content. Therefore, for clarity in the title of Figure 3 has been written the sentence "The 100% of thiamine content corresponds to the after-cooking thiamine content." For the legend of has been added

Question:

Line 260: Describe the freezing procedure in the “material and methods” section.

Answer:  The freezing procedure has been described in the “Material and methods” section.

Has been written in  2.1.2. Gnocchi dumplings formulations -   “Next, all the variants of gnocchi were divided, into two parts, for:

slow freezing at 12 hours for−21°C (in a domestic freezer).” and in 2.1.3.2. Gnocchi dumplings with preparations impregnated “The uncooked gnocchi samples were stored at −21°C for 90 days in vacuum-sealed, medium density polyethylene bags. Storage for the time had the least effect sensory quality of this product (own unpublished data).”

Question:

Line 264: Why are not significant the losses?? Almost 25% of the initial thiamine content disappear (according to Figure 3).

Answer:  Thiamine is one of the most labile vitamins. This is confirmed by previous studies related to both high-temperature processing and storage. In the Discussion has been written: During similar heating conditions of meat products, thiamine losses amounted to 50%.

Has been added “The earlier study indicates that the activation energy, in rosehip nectar, for thiamine is lower than  L-ascorbic acid and riboflavin. It was found that between 70 to 95 0C were found to be 36.38, 55.30, and 37.15 kJ/mol, respectively.” “The analysis of Japanese meals have shown losses of thiamin after cooking in average 50%. In these studies was found that the cooking losses of thiamin were particularly large in rice and green vegetables.  Also, high losses of thiamine (45%) were observed in the samples of fortified with thiamine rice and cooking in a microwave oven. It was also found that he content of thiamine was reduced by 50-70% for enriched polished and parboiled rice. “

Question:

Line 267: The results presented not match with the T1/2, which is the focus of the development of the results.

Answer:  The results of the T1/2 has been corrected. The errors of writing have been corrected.

Question:

Figure 3: The experimental points do not be connected, add the fitted line and the abscise units. The graphic need to be improved in size, and the colors used made reading difficult. The results did not match with the results from Figure 2.

Answer:   The table has not been changed.

Question:

Table 2: Add the variation interval (±), add the slop (a) and intercept (b) significant figures. Why the T1/2 were extrapolated? (The storage days were not enough to find the real T1/2?), and the extrapolation was not the best option.

Answer:  The variation interval (±), add the slop (a) and intercept (b) significant figures have been added. The results of the T1/2 has been corrected. The errors of writing have been corrected.

Question:

Table 3. Consider the significant figures according to the variation interval.

Answer: The table has not been changed.

Question:

Line 359: The cited experiment is yours or from the cited authors (48-50)?

Answer:  The cited experiment is from the cited authors (48-50). The sentence “ In this experiment, the application of thermal processing in a convection oven, with steam as a heating medium, retained higher amounts of thiamine in the gnocchi.” has been corrected to “In our experiment, the application of thermal processing in a convection oven, with steam as a heating medium, retained higher amounts of thiamine in the gnocchi.” and has been written from a new verse.

Question:

Line 360: Describe in MM the “convection oven” used in this study.

Answer:  The “convection oven” has been described in “Material and methods” section: “Cooked - The gnocchi was steamed in a convection oven in a convection oven (Rational, Landsberg am Lech, Germany), with steam as a heating medium and using boiling tap water (initial water content - 1:2 (m/v)) for 5 min at 1000C.”

Question:

Line: 371-372: In this study, the antioxidant activity was not evaluated, therefore, it is impossible to find a “good” correlation between antioxidant activity and thiamine stability.

Answer:   The sentence “Research results showed a strong positive correlation between the antioxidant activity and stability of thiamine.”  has  been deleted.  The sentence “Further research on correlation between the antioxidant activity of the analyzed carriers and stability of thiamine is necessary to clarify this point.” has been added after: “The earlier studies showed that the antioxidant activity of the pumpkin products was significantly higher than for products with broccoli and cauliflower [52–54].”

Round 2

Reviewer 2 Report

The authors addressed adequately the reviewers' comments and the manuscript can be accepted for publication, after careful revision of English language.

Author Response

Poznań, 01 March 2021

Dear Reviewer,

Reference: ID:  foods-1102668 entitled "Pumpkin, cauliflower and broccoli as new carriers of thiamine compounds for food fortification”

The publication has been corrected with careful revision of English language.

We hope that after these corrections you will find our paper suitable to publish in Foods Journal.

KRYSTYNA SZYMANDERA-BUSZKA

The corresponding author

Reviewer 3 Report

Line 165: Add the soaking lapse-time

Line 296: Maybe is better to define the “degradation kinetic constants” than the “reaction rate constants”

Line 297: Explain the mechanisms for the first-order kinetic model

Line 326: It would be interesting to start explaining the differences in the thiamine content of the different vegetable carriers regarding the initial content in the raw vegetables (without soaking-controls). Depending on the carrier, the adsorption of the different types of thiamine in the surface of the vegetables may be different due to the active points from the food matrix, and consequently, their initial content can vary. It would be interesting to explain if the thiamine deposition is similar or not and if it affects the final thiamine content during the storage and also in the gnocchis.

Author Response

Poznań, 01 March 2021

Dear Reviewer,

Reference: ID:  foods-1102668 entitled " Pumpkin, cauliflower and broccoli as new carriers of thiamine compounds for food fortification”

We would like to thank you for your review of our paper. The answer was given to all the comments.

The answers to comments and suggestions to the author's amendments were introduced to work, as suggested - green marked.

The publication has been corrected with careful revision of English language.

We hope that after these corrections you will find our paper suitable to publish in Foods Journal.

KRYSTYNA SZYMANDERA-BUSZKA

The corresponding author

         Question:

Line 165: Add the soaking lapse-time

Answer: Has been corrected “The impregnation process of vegetables consisted in their soaking in a thiamine……” to “The impregnation process of vegetables consisted in their soaking (10 min) in a thiamine………..”

Question:

Line 296: Maybe is better to define the “degradation kinetic constants” than the “reaction rate constants”

Answer:   Has been corrected

Question:

Line 297: Explain the mechanisms for the first-order kinetic model

AnswerThese sentences “Degradation kinetic constants were determined by fitting the experimental data to first-order kinetic model and modelled as a function of storage time.” has been deleted because these data are not converted and presented in this publication.

Question:

Line 326: It would be interesting to start explaining the differences in the thiamine content of the different vegetable carriers regarding the initial content in the raw vegetables (without soaking-controls). Depending on the carrier, the adsorption of the different types of thiamine in the surface of the vegetables may be different due to the active points from the food matrix, and consequently, their initial content can vary. It would be interesting to explain if the thiamine deposition is similar or not and if it affects the final thiamine content during the storage and also in the gnocchis.

Answer: The sentences have been added:

The analysis of the relation of thiamine content in carriers to the initial content in the vegetables showed the effectiveness at the level of 75 - 89%. The highest effectiveness of thiamine fortification was found for muscat pumpkin (89%). The effectiveness depended also on the forms of thiamine. The results showed lower content of thiamine pirophosphorane than hydrochloride compared to the initial content in the vegetables for hokkaido pumpkin (76%) and broccoli (77%).
